# Defucosylation of Tumor-Specific Humanized Anti-MUC1 Monoclonal Antibody Enhances NK Cell-Mediated Anti-Tumor Cell Cytotoxicity

**DOI:** 10.3390/cancers13112579

**Published:** 2021-05-25

**Authors:** Ying Gong, Roel G. J. Klein Wolterink, Valeriia Gulaia, Silvie Cloosen, Femke A. I. Ehlers, Lotte Wieten, Yvo F. Graus, Gerard M. J. Bos, Wilfred T. V. Germeraad

**Affiliations:** 1Department of Internal Medicine, Division of Hematology, Maastricht University Medical Center+, 6229 HX Maastricht, The Netherlands; y.gong@maastrichtuniversity.nl (Y.G.); roel.klein.wolterink@gmail.com (R.G.J.K.W.); gulaya.lera@gmail.com (V.G.); f.ehlers@maastrichtuniversity.nl (F.A.I.E.); gerard.bos@mumc.nl (G.M.J.B.); 2GROW—School for Oncology and Developmental Biology, Maastricht University, 6229 GT Maastricht, The Netherlands; l.wieten@mumc.nl; 3Champalimaud Research, Champalimaud Centre for the Unknown, 1400-038 Lisbon, Portugal; 4CiMaas BV, 6229 EV Maastricht, The Netherlands; cloosensilvie@hotmail.com (S.C.); y.graus@cimaas.com (Y.F.G.); 5Department of Transplantation Immunology, Tissue Typing Laboratory, Maastricht University Medical Center+, 6229 HX Maastricht, The Netherlands

**Keywords:** antibody therapy, natural killer cells, MUC1, antibody-dependent cellular cytotoxicity, breast cancer

## Abstract

**Simple Summary:**

Antibodies with their high specificity to antigens have been widely used in cancer immunotherapy. Natural killer (NK) cells are a group of innate immune cells which have strong cytotoxicity against cancerous cells, virus infected cells, or transformed cells. NK cells express abundant Fc receptors that can bind tumor-specific antibodies, thus allowing them to precisely redirect and eliminate cancer cells. In this study, we demonstrated that NK cells cytotoxicity toward MUC1-positive hematologic and solid tumor can be further enhanced by a humanized 5E5 anti-MUC1 antibody. Furthermore, Fc defucosylation of the antibodies further boosted the kill capacity of NK cells. We believe that our humanized anti-MUC1 antibody is a promising therapeutic candidate for clinical cancer treatment.

**Abstract:**

Antibodies are commonly used in cancer immunotherapy because of their high specificity for tumor-associated antigens. The binding of antibodies can have direct effects on tumor cells but also engages natural killer (NK) cells via their Fc receptor. Mucin 1 (MUC1) is a highly glycosylated protein expressed in normal epithelial cells, while the under-glycosylated MUC1 epitope (MUC1-Tn/STn) is only expressed on malignant cells, making it an interesting diagnostic and therapeutic target. Several anti-MUC1 antibodies have been tested for therapeutic applications in solid tumors thus far without clinical success. Herein, we describe the generation of fully humanized antibodies based on the murine 5E5 antibody, targeting the tumor-specific MUC1-Tn/STn epitope. We confirmed that these antibodies specifically recognize tumor-associated MUC1 epitopes and can activate human NK cells in vitro. Defucosylation of these newly developed anti-MUC1 antibodies further enhanced antigen-dependent cellular cytotoxicity (ADCC) mediated by NK cells. We show that endocytosis inhibitors augment the availability of MUC1-Tn/STn epitopes on tumor cells but do not further enhance ADCC in NK cells. Collectively, this study describes novel fully humanized anti-MUC1 antibodies that, especially after defucosylation, are promising therapeutic candidates for cellular immunotherapy.

## 1. Introduction

Cancer is still one of the leading causes of death around the globe. One hallmark of this neoplastic disease is avoidance of immune destruction. In hematologic malignancies, the tumor cells are dispersed in the blood, bone marrow, and lymph nodes, which in principle facilitates access to tumors by immune cells to conduct eradication [1]. However, both in solid and hematological tumors, malignant cells aggregate and cultivate in a suppressive micro-environment, including hypoxia, low pH and inhibitory cytokines, and molecules, which is detrimental to the function of immune cells [2]. Furthermore, malignant cells induce immune dysregulation by down-regulation of ligands that are natural activators of immune cells [3]. Therefore, tumor immunotherapy comprises various techniques that aim to enhance intrinsic immune mechanisms to promote eradication of tumor cells. Given the remarkable progress made with tumor immunotherapy, it was recently identified as a major breakthrough in clinical cancer treatment [2].

One of the most common forms of immunotherapy is the use of monoclonal antibodies (mAbs), designed to precisely influence the host response to tumor cells [4]. This treatment has a high specificity for tumor cells and thus has little or no side effects on the normal tissue [5]. The functional part of mAbs includes the antigen-binding F(ab)2′ part and an Fc region. The antigen-binding fragment determines the specificity of mAb. The Fc portion of the mAb bind on the Fc receptor expressed on the effector immune cells, including natural killer (NK) cells, macrophages, and dendritic cells [6].

NK cells belong to a group of innate immune cells that make up around 10% of the peripheral blood mononuclear cells (PBMC). NK cells are sentinels for “missing-self” cells and are able to distinguish virally infected cells and tumor cells from healthy counterparts via germline-encoded activating and inhibitory receptors. The activating receptors expressed on NK cells can identify appropriate ligands on virally infected, tumor, senescent, and stressed cells [7]. These ligands will deliver an activating signal to NK cells, thereby initiating cytotoxic processes [8]. In contrast, the inhibitory receptors suppress NK cell activation and subsequently prevent killing of normal, healthy cells through recognition of self-proteins such as the widely-expressed major histocompatibility complex class I molecules (MHC-I) [9]. Moreover, human NK cells express the FcγRIIIa receptor (also known as CD16a), which recognizes the Fc fragment of IgG. This induces NK cell-mediated eradication of IgG-opsonized abnormal cells via the secretion of their cytotoxic granules in a process called antibody-dependent cellular cytotoxicity (ADCC) [6]. Modulation of glycosylation of the antibody’s Fc tail influences the interaction with the FcγRIIIa receptor and can modulate this NK cell effector function. [10] More specifically, removal of fucose on the N-glycan of the Fc tail has been shown to increase Fc-binding affinity to the FcγRIIIa, leading to enhanced ADCC [11,12].

Mucin1 (MUC1) is a highly glycosylated transmembrane protein expressed on the apical side of the cell membrane, which plays a paramount role in the protection and lubrication of normal epithelial cells [13]. In normal cells, the peptide core in heavily glycosylated MUC1 is masked by the O-glycan moieties that protect MUC1 from proteolytic cleavage enzymes. In addition, the glycosylation of MUC1 also stabilizes mucins on the apical side of the membrane by interfering with clathrin-mediated endocytosis. Cancer-related MUC1 proteins have shorter and less dense O-glycan side chains, resulting in exposure of the core domains of the protein on the cell surface. The MUC1 under-glycosylation results in exposure of the epitopes MUC1-Tn and MUC1-STn to the immune system. This feature allows for the design of antibodies that can distinguish between tumor and normal cells [14]. Therefore, MUC1 has been investigated as a promising target toward solid tumors and even placed second as most favorable targets in immunotherapy [15]. Numerous groups have developed monoclonal antibodies against MUC1 both for diagnostic and therapeutic applications [16]. One example is the murine 5E5 antibody that recognizes MUC1-Tn and MUC1-STn epitopes [17]. Several groups have also successfully generated promising chimeric antigen receptor T cells (CAR-T) based on the various murine mAb antibodies sequences [18]. Furthermore, MUC1 conserved peptide vaccine or mRNA-pulsed dendritic cells have also been used to induce immune responses against MUC1-positive tumors [19]. However, clinical studies using MUC1 antibodies have so far not resulted in the identification of a potent anti-MUC1 antibody that can be used for cancer treatment. Therefore, we hypothesized that a humanized version of a cancer-specific murine anti-MUC1 antibody can be used to activate NK cells for cancer immunotherapy. We also asked if defucosylation of the Fc tail of this antibody further enhanced anti-tumor responses.

In this study, we explored the function of the humanized 5E5 mAb in its natural IgG configuration and compared it to the defucosylated IgG, aiming to increase antibody-dependent cellular cytotoxicity by NK cells. We found that humanized anti-MUC1 antibodies indeed promoted ADCC and that defucosylated antibodies performed better. As the murine 5E5 antibody has been demonstrated as an attractive mAb candidate [20], humanized, defucosylated 5E5 could become a potential therapeutic mAb in many epithelial cancers.

## 2. Materials and Methods

### 2.1. Antibodies

The original murine anti-MUC1 monoclonal antibody 5E5 (murine IgG1, binding MUC1 Tn/STn epitopes) was kindly provided by Professor Henrik Clausen (University of Copenhagen, Denmark). In addition, 214D4 (murine IgG1; binding pan-MUC1 epitopes) was kindly provided by Dr. John Hilkens (The Netherlands Cancer Institute, Amsterdam, The Netherlands).

CIM301-1 mAb is a recombinant fully human IgG1 comprising the 5E5 humanized VH and VL amino acid sequence. These human sequences were determined by investigators affiliated to the Cancer Research Technology Ltd. (London, UK) and were obtained under a research license by CiMaas. The whole corresponding coding sequence was first designed in silico; subsequent DNA was synthesized by Eurogentec (Liège, Belgium); and the cloning of the appropriate expression vectors was performed at GeneArt (Thermo Fisher Scientific, Regensburg, Germany). CIM301-4 (anti-HIV gp120) is a recombinant human IgG1 recognizing an epitope overlapping the CD4-binding site of gp120. CIM301-1 and CIM301-4 mAb were produced in CHO-K1 (GeneArt). CIM301-8 is a recombinant human IgG1 comprising the same 5E5 humanized VH and VL amino acid sequence as CIM301-1 but with an ADCC optimized non-core-fucosylated Fc-domain based on Lonza Potelligent^®^ technology produced transiently in CHO-K1SV (Lonza, Basel, Switzerland) on a research license under the Biowa technology to CiMaas. Antibodies used in this study are summarized in Table 1, and their structures are depicted in Figure 1A.

### 2.2. Cell Lines and Cell Cultures

CHO *ldlD* cells were transfected with the coding sequence of MUC1 protein to produce CHO *ldlD* MUC1 cells. Cells were maintained in IMDM medium (Thermo Fisher Scientific, Waltham, MA, USA) with 10% FCS supplement with Gentamicin 418 (Thermo Fisher Scientific) at a concentration of 0.5 mg/mL. To induce the MUC1-Tn epitope on CHO-*ldlD*-MUC1 cells, 1 μM of *N*-acetylgalactosamine (GalNAc) (Sigma-Aldrich, Munich, Germany) was added to the medium, as described previously [21]. T-47D cells (HTB-133, ATCC, Manassas, VA, USA) were maintained in RPMI1640 (Thermo Fisher Scientific, Waltham, MA, USA) supplemented with 10% FCS (Greiner Bio-One, Frickenhausen, Germany), 1% penicillin-streptomycin (Thermo Fisher Scientific, Waltham, MA, USA), and 0.2 U/mL bovine insulin (Sigma, Munich, Germany). MCF7 cells (HTB-22, ATCC, Manassas, VA, USA) were cultured in EMEM (ATCC, Manassas, VA, USA) with 10% FCS, 1% penicillin-streptomycin and 10 µg/mL human recombinant insulin (Sigma, Munich, Germany). SK-BR-3 (ACC 736, DSMZ, Braunschweig, Germany) cells were cultured in McCoy’s 5A (Thermo Fisher Scientific, Waltham, MA, USA) culture medium (Invitrogen, Carlsbad, CA, USA), supplemented with 20% FCS and 1% penicillin-streptomycin. K-562 cells (CCL-243, ATCC, Manassas, VA, USA) were maintained in IMDM medium (Thermo Fisher Scientific, Waltham, MA, USA) with 10% FCS and 1% penicillin-streptomycin. Jurkat cells (ACC 282, DSMZ, Braunschweig, Germany) were cultured in RPMI 1640 medium with Glutamax (Thermo Fisher Scientific, Waltham, MA, USA), 10% FCS and 1% penicillin-streptomycin. All these cell lines were purchased as indicated between parenthesis followed by the generation of master cell and working cell banks. Cells were used from the working cell banks up to passage 10.

### 2.3. Human NK Cell Isolation and Activation

Primary human NK cells were isolated from anonymous buffy coats (Sanquin, Maastricht, The Netherlands). The use of buffy coats, being a by-product of a required Medical Ethical Review Committee (METC) procedure, does not need ethical approval in The Netherlands under the Dutch Code for Proper Secondary Use of Human Tissue. NK cells were isolated by negative selection with an NK cell isolation kit (Miltenyi Biotec, Bergisch Gladbach, Germany) using MACS beads, as previously described [22]. Average purities were >95%. NK cells were cultured in RPMI-1640 medium (Thermo Fisher Scientific, Waltham, MA, USA) supplemented with 10% FCS, 1% penicillin-streptomycin, and 1000 IU/mL recombinant human IL-2 (Proleukin, Novartis, Basel, Switzerland).

### 2.4. Flow Cytometry

For flow cytometric analysis, cells were harvested, washed with PBS (Sigma, Munich, Germany), and first stained with Live/Dead Fixable Aqua Dead Cell Stain Kit (Thermo Fisher Scientific, Waltham, MA, USA) in PBS on ice for 30 min. Then, 0.5 × 10^6^ CHO cells and tumor cells were resuspended in 100 μL PBS and stained with 1 μg/mL murine mAb (5E5 or 214D4) or 1 μg/mL (or other concentrations, as indicated) humanized mAb (CIM301-4, CIM301-1 and CIM301-8). After washing twice with PBS, cells were stained with secondary antibodies. For murine mAb, primary antibodies were detected with 0.5 μg/mL AlexaFluor 647-conjugated goat anti-mouse IgG (H + L) (Jackson Immuno Research, Cambridgeshire, UK). CIM301-4, CIM301-1, and CIM301-8 mAbs were detected with 0.5 μg/mL AlexaFluor 647-conjugated goat anti-human IgG (H + L) (Jackson Immuno Research, Cambridgeshire, UK). Cells were washed twice with PBS after a 15 min incubation with secondary mAb. Cell pellets were resuspended in 200 µL PBS for flow cytometric analysis. Fluorescence was read on a CantoII flow cytometer (BD Biosciences, San Jose, CA, USA). Data were analyzed with FlowJo version 10.7 (TreeStar, Ashland, OR, USA) software.

### 2.5. NK Cell Degranulation Assay

To evaluate NK cell activation by tumor cells or mAbs, CD107a expression on NK cells was analyzed using flow cytometry as previously described [22]. VioBlue-labeled anti-CD107a (clone H4A3, Miltenyi Biotec, Bergisch Gladbach, Germany) or the corresponding isotype was added to the wells immediately after combining 10^5^ NK cells with 10^5^ cancer cells with or without the various humanized anti-MUC1 mAbs in a 96-well plate. After 1 h of incubation at 37 °C in humidified air containing 5% CO_2_, 10 μg/mL Brefeldin A (BFA, BD Bioscience, San Jose, CA, USA) was added to each well. After 3 h of further incubation, plates were put on ice and washed with PBS. After centrifugation, the supernatant was discarded, and 50 µL of antibody mix was added to each well: anti-CD3-FITC (SK7, Miltenyi Biotec, Bergisch Gladbach, Germany), anti-CD56-PerCP-Vio770 (REA196, Miltenyi Biotec, Bergisch Gladbach, Germany), and anti-CD16-APC-H7 (3G8, BD, San Jose, CA, USA).

### 2.6. NK Cell Cytotoxicity (ADCC) Assay

NK cells isolated using negative selection as described above were used in cytotoxicity assays. T-47D and Jurkat cells were used as targets to investigate NK cell killing capacity. The antibodies CIM301-1 and CIM301-8 were added to evaluate their ability to enhance NK cell-mediated target-cell killing. Target cells were labeled with Cell Tracker Deep Red Dye according to the manufacturer’s protocol (Thermo Fisher Scientific, Waltham, MA, USA) the night before the cytotoxicity assay. Tumor cells were harvested using trypsinization and washing and were seeded at 2 × 10^4^ cells per well in round-bottom 96-well plates. Then, NK cells were added at various effector: target (E:T) ratios. At the same time, different dilutions of anti-MUC1 antibodies were added. The total culture volume was 200 μL per well. After 30 min preincubation with mAb, NK cells were added at various effector: target (E:T) ratios. After 4 h of incubation, cells were put on ice and stained with Live/Dead Fixable Aqua (LDA) Dead Cell Stain Kit (Thermo Fisher Scientific, Waltham, CA, USA). The percentage of specific killing was calculated using the following formula:(1)$$\%\;specific\;killing=\frac{\%\;LDA\;positive\;target\;cells-\%spontaneus\;LDA\;positive\;target\;cells}{\%\;vital\;cells}\times 100$$

### 2.7. Endocytosis Inhibitors in Degranulation and Cytotoxicity Assays

Prochlorperazine (PCZ, Sigma, Munich, Germany) and Dyngo4A (Abcam, Cambridge, UK) were resuspended in 0.1% (*v/v*) DMSO (Sigma, Munich, Germany), which was also used as a solvent control. Degranulation and cytotoxicity assays were performed as described above, with addition of endocytosis inhibitors to the co-culture during the last hour of the assays in concentrations of 5 μM PCZ and 30 μM Dyngo4A.

### 2.8. Statistical Analysis

All statistical tests used in this study were completed with GraphPad Prism 8 software (Graphpad Software, San Diego, CA, USA). The specific statistical tests used for each comparison are specifically annotated in the figure legends, respectively.

## 3. Results

### 3.1. Fully Humanized Anti-MUC1 Antibodies Specifically Recognize Tumor-Associated MUC1 Glyco-Epitopes

To further explore the therapeutic potential of anti-MUC1 antibodies for cancer immunotherapy, we generated fully humanized anti-MUC1 antibodies based on the 5E5 murine antibody (Figure 1A). The 5E5 antibody was previously shown to specifically recognize cancer-specific MUC1 epitopes (MUC1-Tn/STn) and elicits strong immune responses in mice [17]. In addition to a fully humanized 5E5 antibody (designated CIM301-1), we generated CIM301-8, a defucosylated variant of the antibody with the aim of optimizing antibody-dependent cellular cytotoxicity (ADCC). Lastly, we generated a control antibody, CIM301-4, directed against the non-relevant HIV-gp120 epitope (Table 1).

We first confirmed that, in agreement with our previous data, the murine anti-MUC1 antibodies 214D4 (pan-MUC1) and 5E5 (cancer-specific MUC1-Tn/STn) recognized the relevant MUC1-Tn/STn epitopes on CHO cells expressing MUC1 epitopes [21], while no binding was observed in the parental CHO *ldlD* cell line lacking MUC1 epitopes (Figure 1B). Likewise, the newly generated humanized anti-MUC1 antibodies CIM301-1 and CIM301-8 specifically binded to CHO cell lines expressing MUC1-Tn/STn epitopes, while the control antibody CIM301-4 showed no binding (Figure 1C). Moreover, CIM301-1 and CIM301-8 antibodies showed highly preferential binding to CHO cell lines expressing cancer-related MUC1-Tn/STn epitopes over CHO *ldlD* cell lines expressing non-modified MUC1. Next, we analyzed whether the humanized anti-MUC1 antibodies also recognized MUC1 Tn epitopes expressed on cancer cell lines. Previous studies have shown that the Jurkat cell line strongly expresses MUC1 Tn antigens due to a mutation in the *COSMC* gene that interferes with protein glycosylation [23]. Indeed, compared with the MUC1 Tn epitope-negative K-562 cell line, murine (Figure 1D) and humanized (Figure 1E) anti-MUC1 Tn antibodies displayed high binding affinity for Jurkat cells. Along the same line, the MUC1-expressing breast cancer cell lines MCF7 and T-47D also showed strong staining with anti-MUC1 Tn antibodies, while the MUC1 Tn-negative breast cancer cell line SK-BR-3 was only recognized by pan-MUC1 antibodies (Figure 1D). Together, these results confirm that the humanized anti-MUC1 antibodies CIM301-1 and CIM301-8 specifically recognize the cancer-associated MUC1 Tn epitope. Therefore, we explored the applicability of these antibodies for immunotherapy in combination with NK cells.

### 3.2. Humanized Anti-MUC1 Antibodies Conjugate NK Cells and Induce Degranulation

Monoclonal antibodies recognizing tumor antigens can induce ADCC through the binding of the antibody Fc tail to the FcγRIIIa (CD16) molecules on NK cells [6]. Therefore, we asked whether the humanized CIM-301 anti-MUC1 antibodies can indeed bind to NK cells (gating strategy in Appendix A) by detecting labeled antibodies bound to the Fc portion of the anti-MUC1 antibody (Figure 2A). As expected, the murine anti-MUC1 antibodies did not bind to human NK cells (Figure 2B), while the fully humanized CIM301-1 and CIM301-8 antibodies showed robust binding to NK cells (Figure 2C). Notably, the defucosylated CIM301-8 displayed significantly stronger binding to Fc receptors on NK cells (Figure 2C,D), confirming that removal of oligosaccharides in the Fc region of the antibody could be beneficial for ADCC in NK cells [6]. Activation of NK cells by cross-linking CD16 with antibodies induces strong activation in NK cells without the need for other activation signals [24]. NK cell activation leads to degranulation, which can be measured as CD107a expression levels. Therefore, we determined whether incubation of NK cells with humanized anti-MUC1 antibodies induced CD107a expression (Figure 2E, with gating strategy in Appendix A). Indeed, we found that incubation of primary NK cells with humanized anti-MUC1 only (without tumor cells) could induce degranulation (Figure 2F,G). Compared with the CIM301-4 control antibody, the defucosylated anti-MUC1 antibody CIM301-8 showed higher induction of NK cell activation. Thus, humanized anti-MUC1 antibodies bind to the FcγRIIIa receptor on human NK cells and can induce their activation, potentiating them to recognize and lyse MUC1-expressing tumor cells.

### 3.3. Increasing Concentrations of Humanized Anti-MUC1 Antibodies Enhance NK Cell Activation and Induce CD16 Down-Regulation, but Do Not Further Enhance Tumor Cell Killing

Before comparing the standard humanized anti-MUC1 antibody to its defucosylated counterpart, we first determined the optimal antibody concentration for in vitro use. Monoclonal antibodies for cancer immunotherapy exert direct effects on the tumor cells, the effector NK cells, and on NK cell-mediated cytotoxicity. Therefore, we tested the defucosylated CIM301-8 antibody in all of these conditions. First, we incubated Jurkat tumor cells alone with increasing concentrations of humanized anti-MUC1 (CIM301-8) antibody and determined the level of saturation of the MUC1 epitopes and direct toxicity to tumor cells (Appendix A). We observed that binding of MUC1 epitopes (Appendix A) and direct cytotoxic effects on the tumor cells (Appendix A) were both dose-dependent. We then tested whether increasing antibody concentrations promoted antibody binding by NK cells and their degranulation (Figure 3A). Antibody-mediated NK cell activation via CD16 is a strong inducer of degranulation, but also induces down-modulation of CD16, most likely to prevent overactivation of NK cells [24,25]. In line with these results, we observed that increasing doses of anti-MUC1 antibodies resulted in an increased fraction of degranulating NK cells (Figure 3B). In the presence of anti-MUC1 antibodies, the major proportion of these NK cells were negative for CD16 (Figure 3B and Appendix A) and the overall CD16 levels decreased with increasing antibody concentrations (Figure 3C and Appendix A). This suggested that the increase in degranulation of NK cells most likely came from cells that were activated by interaction of the antibody with CD16 leading to a loss in CD16 expression on these cells. Not surprisingly, these lower CD16 levels also resulted in lower levels of MUC1 antibodies bound to NK cells (Figure 3D and Appendix A). Importantly, the observed enhanced NK cell degranulation came at the expense of higher NK cell death, almost doubling from 15% at 1 μg/mL to 26.7% at the highest concentration (Figure 3E).

Lastly, we also titrated the antibody concentration in NK cell–Jurkat tumor cell co-culture experiments, using degranulation and specific cytotoxicity as readouts (Figure 4A). The addition of the anti-MUC1 antibody increased both degranulation and cytotoxicity against MUC1 Tn/STn^+^ Jurkat cells (Figure 4B,C; white bars versus red bars). However, we did observe a dose-dependent effect of the antibody-induced degranulation of NK cells (Figure 4B,C; shades of red), but this is not more than the increase that the direct effect of the antibody induced on the NK cells (Figure 3). Instead, the addition of extra NK cells (higher E:T ratios) was the main determinant of tumor cell killing (Figure 4C): in the presence of anti-MUC1 antibodies, the average tumor kill increased with on average 12.8% with every doubling of the E:T ratio. At the lowest E:T ratio tested (0.25:1), on average, 21.1% of the tumor cells were killed, while at the highest E:T ratio (2:1), this was 59.6%.

In summary, the generated defucosylated humanized anti-MUC1 antibody dose-dependently enhanced NK cell degranulation at the expense of CD16 downmodulation and increased NK cell death. NK cell-mediated responses against MUC1+ Jurkat tumor cells were importantly enhanced by the antibody, but antibody concentrations higher than 1 µg/mL showed no additional beneficial effects. Therefore, we concluded that 1 µg/mL is the optimal concentration for further in vitro experiments.

### 3.4. Specific NK Cell-Mediated Anti-Tumor Responses Are Enhanced by Anti-MUC1 Antibodies, Especially after Defucosylation

Next, we asked whether defucosylation of these humanized anti-MUC1 antibodies show more potent anti-tumor responses. We compared the regular anti-MUC1 antibody CIM301-1 to the defucosylated CIM301-8 antibody in degranulation and cytotoxicity assays against two MUC1-Tn+ tumor cell lines (Figure 5A). First, we performed NK cell-tumor cell co-culture experiments and determined whether both antibodies could enhance the activation of NK cells. Indeed, CIM301-1 and CIM301-8 both enhanced NK cell degranulation in co-cultures with Jurkat cells and T-47D cells compared to the control antibody (CIM301-4) (Figure 5B,C). In ADCC assays, both anti-MUC1 antibodies enhanced cytotoxicity against Jurkat (Figure 5D and Appendix A) and T-47D cells (Figure 5E and Appendix A). Compared to the control antibody, the regular antibody increased anti-Jurkat cell responses by 23%, while the defucosylated antibody increased cytotoxicity 49% (Figure 5D). For T-47D, the increases were 16% and 31%, respectively (Figure 5E). These effects were most pronounced at higher E:T ratios and with CIM301-8 antibodies (Appendix A). In summary, these newly generated humanized anti-MUC1 antibodies importantly enhance NK cell-mediated cytotoxicity against different MUC1-Tn/STn epitope-positive tumor cells. In line with results obtained with other antibodies [20,26,27,28], defucosylation of anti-MUC1 antibodies further enhanced anti-tumor responses.

### 3.5. The Endocytosis Inhibitor PCZ Promotes Tumor Antigen Expression but Does Not Enhance Anti-Tumor Responses in NK Cells

Bioavailability of target antigens is an important determinant of the efficacy of antibody-mediated cancer therapy. MUC1 antigen expression on tumor cells is known to be dynamic, due to the internalization of the protein through clathrin- and dynamic-mediated endocytosis [29]. This mechanism may facilitate the escape of tumor cells from NK cell-induced cell death by avoiding antibody opsonization [30]. Endocytosis inhibitors have been demonstrated to augment the ability of immune cells to eradicate EGFR-positive cells in both ex vivo and in vivo models [31]. Therefore, we hypothesized that the endocytosis inhibitors PCZ and Dyngo 4A could be used to increase the bioavailability of MUC1 antigens, thereby further promoting ADCC by NK cells. To test this, we investigated whether the addition of endocytosis inhibitors to tumor cell cultures promoted the expression of tumor-associated MUC1 epitopes (Figure 6A–C). We found that, in the presence of PCZ, MUC1-Tn/STn epitope expression increased two-fold on T-47D cells (Figure 6B) but not on Jurkat cells (Appendix A). However, Dyngo 4A [31], another endocytosis inhibitor, did not alter the expression of MUC1 (Figure 6C and Appendix A). Addition of endocytosis inhibitors to tumor cells in the presence of humanized anti-MUC1 antibodies had no negative effect on tumor cell viability (Figure 6D and Appendix A). However, in co-culture experiments of NK cells and tumor cells (Figure 6E), endocytosis inhibitors neither enhanced NK cell degranulation (Figure 6F and Appendix A) nor ADCC (Figure 6G and Appendix A). Endocytosis inhibitors can thus promote the expression of tumor-associated MUC1 epitopes, but do not have beneficial effects on NK cell-mediated tumor elimination using anti-MUC1 antibodies. Importantly, we observed no negative effects of endocytosis inhibitors on NK cell viability, and we did not observe negative effects of endocytosis inhibitors on cytotoxic capabilities of NK cells (Figure 6G). Alternatively, it is possible that in previous experiments we already attained the maximal stimulation with humanized anti-MUC1 antibodies.

## 4. Discussion

In this study, we investigated the potential of fully humanized anti-MUC1 antibodies based on the murine 5E5 antibody that specifically recognizes MUC1 Tn/STn cancer-associated epitopes. We demonstrated that the newly developed, fully human CIM301-1 and CIM301-8 antibodies also functionally bind to MUC1-Tn/STn epitopes on tumor cell lines, and are capable of enhancing NK cell-mediated cytotoxicity upon binding to CD16. Throughout our analyses, we found that defucosylation of the Fc tail (CIM301-8) further enhanced anti-cancer effects. Together, these results are the next step in the use of these cancer-specific fully human anti-MUC1 antibodies for cancer immunotherapy, especially in the context of adoptive cell therapy.

The capacity of our humanized anti-MUC1 antibodies to enhance cytotoxicity was dependent on the NK cell quantity and on the fucosylation status of the antibody Fc region. In the pooled analysis of ADCC assays with Jurkat cells, fucosylated and defucosylated humanized anti-MUC1 antibodies enhanced cytotoxicity by 23% and 49% (Figure 5D), respectively. ADCC against the breast cancer cell line T-47D was enhanced by 31% by defucosylated antibody CIM301-8 and 16% with CIM301-1 (Figure 5E). These findings are in line with earlier studies investigating enhancement of ADCC by trastuzumab and cetuximab, both currently used in the clinical setting. For instance, the anti-HER2 antibody trastuzumab enhanced ADCC by 21% against the T-47D cell line that expresses HER2 at low to moderate levels [32]. In a direct comparison between fucosylated and defucosylated cetuximab (anti-EGFR), PBMC-mediated ADCC was enhanced by ~30% at various E:T ratios [33]. These findings should be confirmed in patient studies, as in the clinical setting, antibody availability, E:T ratios within the tumor microenvironment, other immune populations, and various external factors greatly impact the antibody’s anti-cancer efficacy.

MUC1 is an interesting target for antibody-based anti-cancer therapy, as it is expressed by a wide variety of tumors, including breast, ovarian, lung, colon, and pancreatic carcinomas as well as multiple myeloma [14]. MUC1 was identified almost 40 years ago, with cancer-specific aberrations in its expression pattern and glycosylation being recognized later [34]. The identification of the cancer-associated Tn and STn glycoforms of MUC1 also lead to the introduction of the murine 5E5 antibody [27]. There are several examples of MUC1 antibodies, such as HMFG1, HMFG2 [35], and SM3 [36], targeting a broad spectrum of MUC1 epitopes. These have been considered for cancer immunotherapy, but currently without much clinical success. For instance, a humanized form of HMFG1, designated AS1402, was tested in a phase II clinical study that was terminated early because of worse outcomes in patients receiving AS1402 [26]. HMFG2 and SM3 were demonstrated to be effective against MUC1-expressing tumor cell lines in mouse models, both as antibodies and as scFv in the context of a CAR-T [37]. Additionally, ga-tipotuzumab (PankoMab) specifically reacts with cancer-associated MUC1 [28]. Although its safety was confirmed in a phase I trial (ClinicalTrials.gov Identifier: NCT01222624) [38], it did not show beneficial effects in a phase IIb trial in advanced ovarian cancer [39]. The 5E5 antibody under investigation in this study is of particular interest, because of its specificity for Tn/STn epitopes. It preferentially recognizes the Tn- and STn-carrying GSTA region of MUC1, while PankoMab also recognizes the less cancer-specific T-carrying epitopes on the PDTR region [17,27]. Given the wide expression pattern of MUC1, the narrowed specificity of the 5E5 antibody is an advantage for any clinical application, as it reduces the risk of recognition of non-cancerous epitopes.

NK cells efficiently eliminate tumor cells and pathogens after binding the Fc portion of the monoclonal antibody, mainly through FcγRIIIa (CD16a) [6]. Defucosylation of the Fc tail can be used to promote the binding to effector cells and has shown encouraging results in in vitro experiments [40]. In line with this, we also found that the glycoengineered, defucosylated variant of the humanized anti-MUC1 antibody (CIM301-8) outperformed CIM301-1. Currently, three defucosylated antibodies are used for clinical care, and many more are being evaluated in clinical trials. Obinutuzumab is an anti-CD20 defucosylated mAb approved by the FDA in 2013 and is used for the treatment of patients suffering from follicular lymphoma and CLL [41]. Compared to its fucosylated counterpart rituximab, the most widely used monoclonal antibody in the clinic, obinutuzumab enhances ADCC of NK cells in human lymphoma xenograft models and displays superior anti-tumor activity [42]. The other two defucosylated mAbs approved for clinical application are mogamulizumab, an anti-CCR4 antibody used for T cell lymphoma [43], and benralizumab, an anti-IL-5R antibody used for severe eosinophilic asthma [44]. In contrast to obinutuzumab, the latter two antibodies have not been directly compared to fucosylated antibodies. The previously mentioned anti-MUC1 antibody gatipotuzumab is defucosylated and has been tested in two clinical trials, so far without positive results [39]. Currently, another study investigates the combination of gatipotuzumab with tomuzotuximab (anti-EGFR) for treatment of patients with metastatic solid tumors (NCT03360734) [45]. In conclusion, antibody defucosylation should always be considered for cancer therapy, as it often enhances binding and clinical effects.

In addition to glycosylation of the Fc tail of an antibody [6,40,46], there are several other determinants of the effectiveness of ADCC mediated by the interaction between NK cells and monoclonal antibodies: the affinity of the monoclonal antibody [47], stability of CD16 expression on NK cells [48], polymorphisms in CD16 [49] and concurrent Toll-like receptor agonists stimulation [50]. Macías-León et al. demonstrated that amino acids H32, A33, H35, H50, S99, T100, and F102 in the heavy chain and Y98 and Y100 in the light chain of the murine 5E5 antibody form the GalNAc-Tn epitope binding domain [51]. Sequence comparison of the humanized key antigen binding amino acids in the complementary determining regions (CDR)-3 show them to be identical to the 5E5 murine antibody [51]. As our humanized 5E5 antibodies bind the MUC1-Tn at the same level with murine 5E5 (Figure 1), we hypothesize that their affinities are similar. Surface plasmon resonance assays with MUC1-Tn peptides and the humanized and murine 5E5 antibodies may validate this hypothesis in the future.

Furthermore, we found that CD16 expression levels decreased with increasing antibody concentrations as reported previously [24,52,53]. This may seem counterproductive: in contrast to other activating receptors, antibody-mediated cross-linking of CD16 alone is sufficient to fully activate NK cells and to trigger degranulation [54]. However, it has been shown that the shedding of CD16 is required to disassemble the established immune synapse between the NK cell and the tumor cell to allow for serial engagement of other targets [52]. Interestingly, in vitro studies have shown that NK cells can eliminate up to seven targets in 12 h [55,56]. In addition, it has been suggested that CD16 down-regulation prevents NK cells from overactivation and exhaustion [57]. Still, lower baseline levels of CD16 correlate with decreased ADCC responses in NK cells: compared with NK cells obtained from healthy donors, it was demonstrated that NK cells from cancer patients showed a significant reduction of both direct killing and ADCC against tumors, which was due to CD16 down-regulation [58]. In cancer patients, the expression levels of CD16, DNAM-1, and NKG2D have also been reported down-regulation on NK cells [59,60]. A Western blot assay using NK cells after stimulation with tumor cells may further add proof to possible CD16 shedding. Most activating receptors on NK cells, such as NKG2D, can at least in part be rapidly recycled. In contrast, down-modulation of CD16 is mediated by proteolytic cleavage by ADAM17 or MMP25 [24,25]. Consequently, recovery of CD16 expression may take days or weeks, as for instance reported after exposure to an influenza vaccine with partial recovery of CD16 expression only at day 18 [61]. Therefore, it is interesting to consider inhibition of proteolytic cleavage of CD16. Indeed, inhibition of the metalloprotease ADAM17 was shown to induce even stronger activation of NK cells [24]. In addition, inhibition of ADAM17 expression in NK cells, for instance using CRISPR/Cas9 or siRNA, could be used to prevent shedding of CD16 on NK cells used for cancer immunotherapy. In line with these results, our anti-MUC1 antibodies induced down-modulation of CD16 expression on NK cells in a dose-dependent manner. While we observed higher activation levels of NK cells with higher anti-MUC1 doses, we also found that NK cell-mediated ADCC responses were not enhanced at antibody concentrations higher than 1 μg/mL. In addition to optimal dosing of antibodies and modulation of CD16 expression, clinical trials were recently launched using adoptive transfer of NK cells expressing the high affinity form of CD16 [48].

In this study, we also investigated whether endocytosis inhibitors can enhance the effects of anti-MUC1 antibodies by preventing the down-regulation of MUC1 epitopes via endocytosis. MUC1 has been demonstrated to interact with the cell membrane through dynamic endocytosis, mediated by clathrin and dynamin [62]. The cell surface expression levels of under-glycosylated, cancer-associated MUC1 are reduced compared with normal MUC1, either due to decreased delivery to the cell membrane or due to faster endocytosis, a critical tumor immune escape mechanism [29]. The process of endocytosis of MUC1 after binding of HMFG1 antibodies in MCF7 cells takes around 15 min [30]. Endocytosis inhibitors such as PCZ and Dyngo 4A have been proven to up-regulate tumor antigens such as EGFR and Her2, thereby enhancing ADCC [31]. In this study, we tested these two endocytosis inhibitors and found that MUC1-Tn/STn expression increased two-fold, though the effect was restricted to PCZ implemented on T-47D cells. There are various possible explanations for this phenomenon. For instance, different clathrin-independent pathways may be responsible for MUC1 internalization upon antibody [31]. Moreover, we found no changes in MUC1-Tn/STn epitope expression levels on Jurkat cells after treatment with either PCZ or Dyngo 4A. This could indicate that the Tn/STn epitope on Jurkat cell is so abundant that regular MUC1 internalization causes only slight changes on the cell surface, leaving the epitope expression pattern virtually unaltered. Thus, the exact mechanisms of MUC1 endocytosis in relation to ADCC need to be further studied to be able to allow anti-MUC1 antibodies to bind optimally to cancer-associated MUC1 epitopes.

## 5. Conclusions

In this study, we demonstrated that the 5E5-based, humanized anti-MUC1 antibodies CIM301-1 and CIM301-8 are potent enhancers of NK cell activation and cytotoxicity against MUC1-Tn/STn positive tumor cells in vitro. Defucosylation (CIM301-8) further potentiated the NK cell response. Hereafter, in vitro binding studies, in vivo animal studies, and clinical trials should be conducted to explore the full therapeutic potential of these newly generated antibodies. Furthermore, the developed antibodies could also be tested in the context of adoptive cell therapy with effector cells redirected with chimeric antigen receptors (CAR) employing the scFv of the humanized 5E5 antibodies. Given the cancer specificity of these 5E5-based antibodies, combined with the fact that many different tumors show expression of cancerous MUC1 epitopes, CIM301-1 but especially CIM301-8 are interesting candidates for cancer immunotherapy.

## Figures and Tables

**Figure 1 cancers-13-02579-f001:**
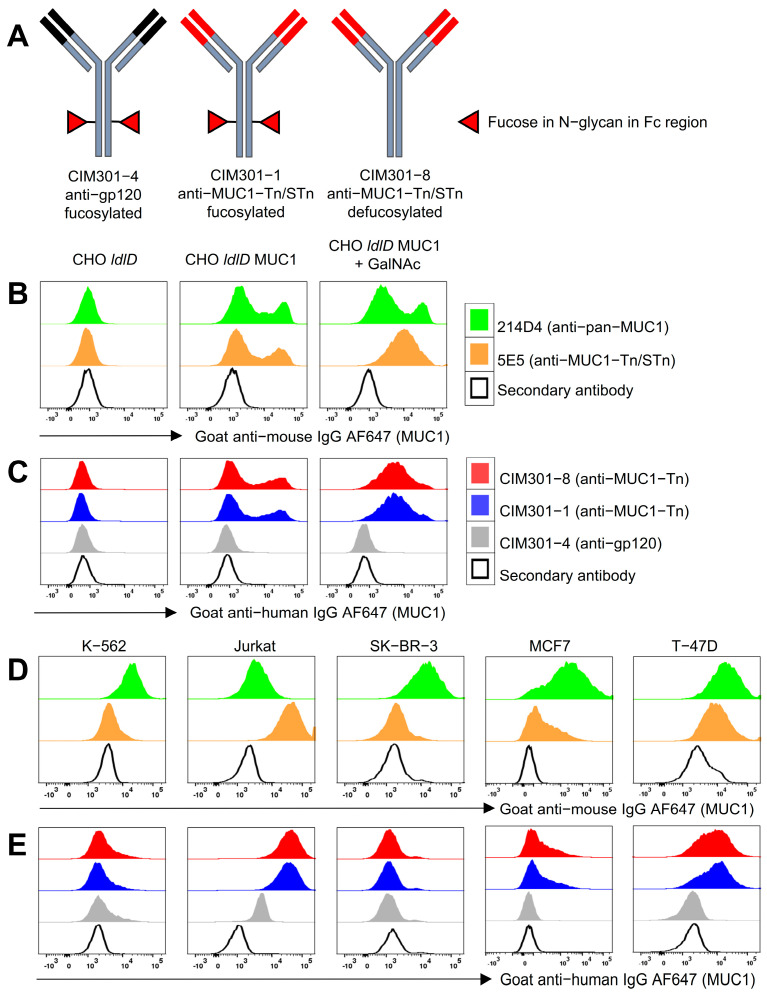
Fully humanized anti-MUC1 antibodies specifically recognize tumor-associated MUC1 epitopes. (**A**) Illustration outlining the design of the antibodies recognizing MUC1 or an irrelevant epitope (control) used in this study. Red triangles indicate glycosylation with fucose of the N-glycans in the antibody Fc tail. (**B**) Overlay histograms of MUC1 expression on CHO cell lines detected using murine antibodies. CHO cells expressed no MUC1 (CHO *ldlD*), glycosylated MUC1 (CHO *ldlD* MUC1), or MUC1-Tn/STn with tumor-associated glyco-epitopes (CHO *ldlD* MUC1 + GalNAc). (**C**) Overlay histograms of MUC1 expression on CHO cell lines detected by humanized antibodies. CHO cell lines as described in (**B**). (**D**) Expression levels of MUC1 or under-glycosylated MUC1 on cancer cell lines detected using 214D4 and 5E5 murine antibodies, respectively. (**E**) Histograms showing binding of humanized anti-MUC1 antibodies to various cancer cell lines.

**Figure 2 cancers-13-02579-f002:**
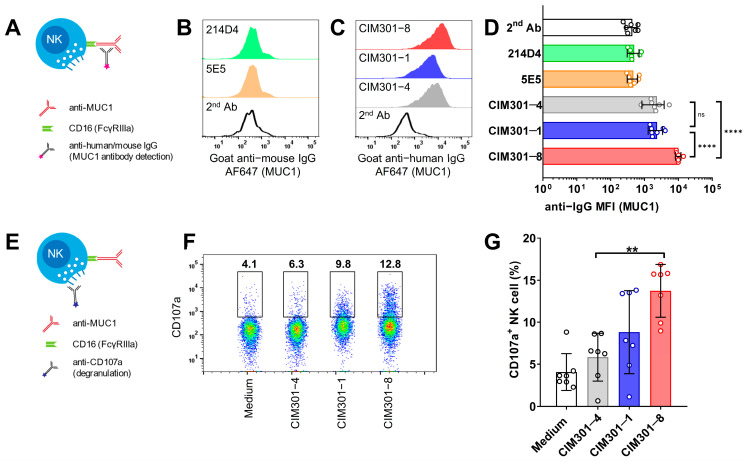
Humanized anti-MUC1 antibodies bind to CD16 on NK cells and induce degranulation. (**A**) Schematic overview outlining the detection of mAb binding via their Fc tail to the FcγRIIIa receptor (CD16) expressed on NK cells. (**B**) Binding of murine anti-MUC1 antibodies on primary human NK cells. Histograms show expression levels of the labeled anti-mouse IgG secondary antibodies used to detect MUC1 antibody binding. (**C**) Binding of humanized anti-MUC1 antibodies on primary NK cells. Experiment as in B, but using humanized anti-MUC1 antibodies, detected using anti-human IgG secondary antibodies. (**D**) Quantification of anti-MUC1 antibody binding to primary human NK cells. (**E**) Scheme depicting analysis of NK cell degranulation. (**F**) Flow cytometric analysis of NK cell degranulation following 4 h incubation of primary human NK cells with anti-MUC1 antibodies. One representative sample is shown with numbers indicating frequencies of cells within the indicated gate. (**G**) Quantification of the degranulation assay shown in (**F**). Bars in (**D**,**G**) show mean ± SD with individual data points as dots. Pooled data from seven independent experiments with different donors performed at different time points. Statistical analysis using one-way ANOVA plus Tukey’s multiple comparisons test. In (**D**), only biologically relevant comparisons were made. Not significant (n.s.); *p* < 0.01 (**); *p* < 0.0001 (****).

**Figure 3 cancers-13-02579-f003:**
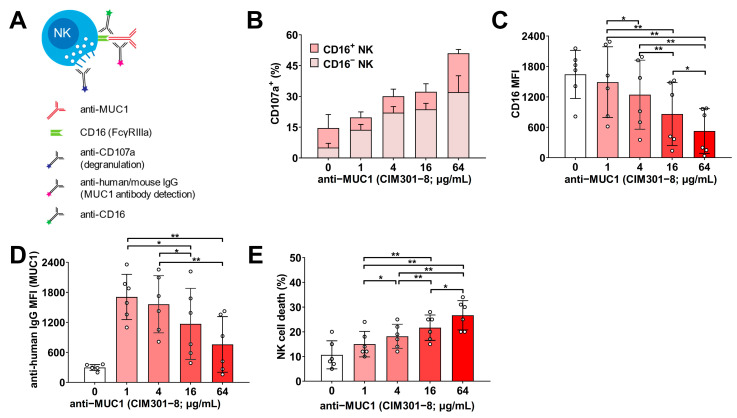
Increasing concentrations of humanized anti-MUC1 antibodies enhance degranulation and induce down-regulation of CD16 expression on NK cells. (**A**) Experimental setup for the detection of anti-MUC1 antibody binding to NK cells, inducing degranulation and down-modulation of CD16 on NK cells. Human NK cells were incubated with increasing concentrations of anti-MUC1 (CIM301-8; defucosylated Fc-tail). Antibody binding to CD16 was analyzed using flow cytometric analysis of anti-human IgG antibodies. (**B**) Stacked bars show the fraction of NK cell degranulation for CD16^+^ and CD16^−^ NK populations in the presence of various antibody concentrations. (**C**) Quantification of CD16 expression on NK cells. (**D**) Quantification of anti-MUC1 antibody binding to the Fc tail of NK cells, shown as median fluorescence index (MFI) of anti-human IgG. (**E**) Quantification of NK cell death after 4 h incubation with increasing doses of anti-MUC1 antibody. Bars indicate mean ± SD, and dots are individual NK cell donors (*n* = 6 per experimental group). Statistical analysis using one-way ANOVA plus Tukey’s multiple comparisons test. *p* < 0.05 (*); *p* < 0.01 (**).

**Figure 4 cancers-13-02579-f004:**
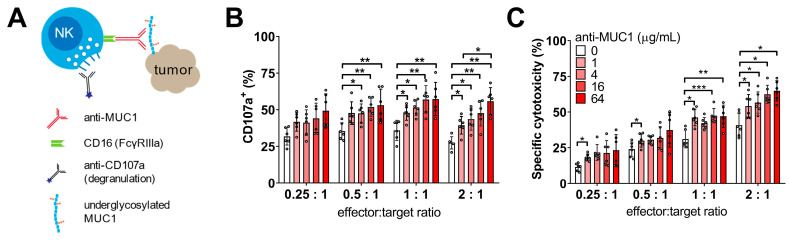
Increasing concentrations of humanized anti-MUC1 antibodies do not further enhance NK cell degranulation and cytotoxicity. (**A**) Experimental overview of the analysis of degranulation and cytotoxicity after co-culture of human NK cells with Jurkat MUC-Tn positive tumor cells in the presence of anti-MUC1 antibodies. (**B**) Fraction of degranulation NK cells after 4 h of co-culture with different effector/tumor ratios and antibody concentrations. (**C**) Quantification of antibody-dependent cellular cytotoxicity at different E:T ratios and antibody concentrations. Bars indicate mean ± SD, and dots are individual NK cell donors from independent experiments (*n* = 6 per experimental group). Statistical analysis using two-way ANOVA plus Tukey’s multiple comparisons test. *p* < 0.05 (*); *p* < 0.01 (**); *p* < 0.001 (***).

**Figure 5 cancers-13-02579-f005:**
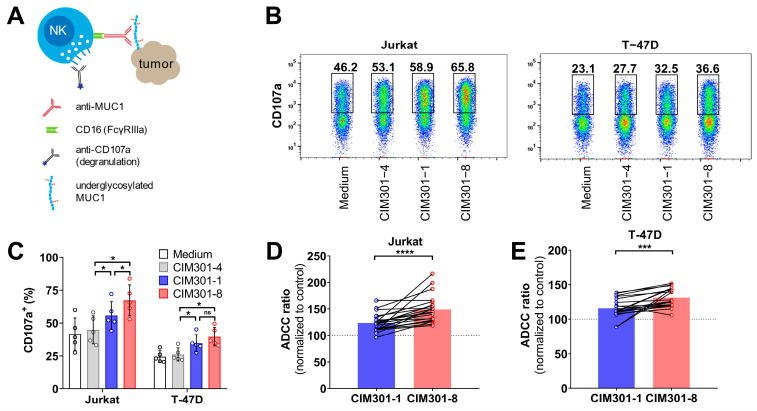
Defucosylation of humanized anti-MUC1 antibodies further enhances specific anti-tumor responses mediated by primary NK cells. (**A**) Scheme outlining the analysis of NK cell degranulation and cytotoxicity in co-cultures of primary human NK cells and MUC1 (S)Tn^+^ tumor cells in the presence of humanized anti-MUC1 antibodies. (**B**) Flow cytometric analysis of NK cell degranulation after 4 h co-culture in the presence of anti-MUC1 antibodies (regular CIM301-1 and defucosylated CIM301-8) or control antibodies (CIM301-4) at 1 μg/mL. Per condition, one representative sample of CD56^+^ NK cells is shown, with gates and numbers indicating the fraction of degranulation NK cells. (**C**) Quantification of the degranulation assays described in (**B**). Pooled data from five independent experiments performed with different NK cells donors at different time points. Differences between groups were tested using repeated measures one-way ANOVA with Šídák’s multiple comparisons test. (**D**) Antibody-dependent cell-mediated cytotoxicity (ADCC) assay using human NK cells co-cultured with Jurkat tumor cells with or without 1 µg/mL regular (CIM301-1) or defucosylated (CIM301-8) antibodies. Dots represent individual data points from six independent experiments with four different effectors: target ratios after normalization to the CIM301-4 control antibody. Dotted line indicates the CIM301-4 baseline level (100%). (**E**) as (**D**), but using T-47D tumor cells as target cells. Differences in ADCC (panels (**D**,**E**)) were calculated using repeated measures one-way ANOVA with Tukey’s multiple comparisons test. *p*-values for comparisons to the control antibody for both antibodies using both tumor cells lines were <0.0001. *p* < 0.05 (*); Not significant (n.s.); *p* < 0.001 (***); *p* < 0.0001 (****).

**Figure 6 cancers-13-02579-f006:**
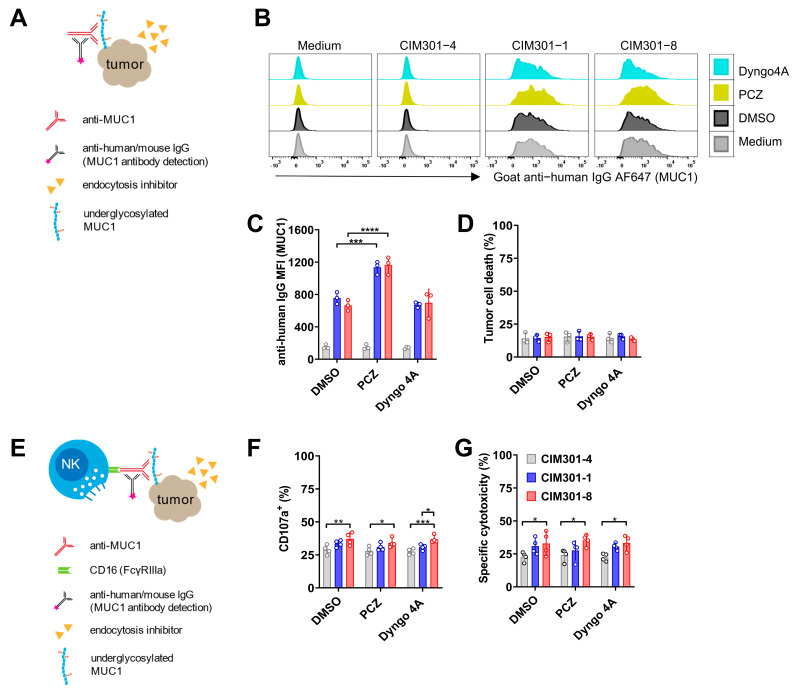
The endocytosis inhibitor PCZ promotes tumor antigen expression on T-47D cells but does not enhance antibody-dependent NK cell-mediated cytotoxicity. (**A**) Experimental setup to determine whether endocytosis inhibitors enhance MUC1 epitope availability on tumor cells. (**B**) Overlay histograms of flow cytometric analysis of MUC1 epitope expression on T-47D tumor cells in the presence of endocytosis inhibitors using regular (CIM301-1) or defucosylated (CIM301-8) anti-MUC1 antibodies or an irrelevant control antibody (CIM301-4). Anti-human IgG antibodies were used to detect antibody binding on tumor cells. Endocytosis inhibitors were dissolved in DMSO, here used as a negative control. One representative sample is shown. (**C**) Flow cytometric quantification of MFI of MUC1 expression levels on T-47D cells after treatment with endocytosis inhibitors as described in (**B**). Differences between control antibody (CIM301-4; grey bars) and anti-MUC1 antibodies (regular CIM301-1 in blue and defucosylated CIM301-8 in red) were all statistically significant with *p* < 0.0001. (**D**) Viability of T-47D tumor cells after incubation with anti-MUC1 antibodies with or without endocytosis inhibitors. Pooled data from three independent experiments performed at different timepoints. (**E**) Schematic overview of assays to test whether endocytosis inhibitors influence NK cell degranulation and cytotoxicity in the presence of anti-MUC1 antibodies. (**F**) Fraction of CD107a+ degranulating human NK cells in co-cultures with T-47D tumor cells at an effector/target ratio of 1:1 in the presence of anti-MUC1 antibodies and endocytosis inhibitors. NK cells and tumor cells were incubated for 4 h, with endocytosis inhibitors (5 μM PCZ, 30 μM Dyngo4A and 0.1% (*v/v*) DMSO) added during the last hour. (**G**) Antibody-dependent NK-cell-mediated cytotoxicity against T-47D tumor cells. Experimental setup as in (**F**). Panels (**F**,**G**) show pooled data from four independent experiments with different donors, performed at different time points. Differences between groups were determined using two-way ANOVA with Tukey’s multiple comparisons test. *p* < 0.05 (*); *p* < 0.01 (**); *p* < 0.001 (***); *p* < 0.0001 (****).

**Table 1 cancers-13-02579-t001:** Properties of anti-MUC1 antibodies.

Name	Description	Isotype	Target
5E5	Mouse antibody anti-MUC1-Tn epitope	Murine IgG1	Muc1-Tn/STn
214D4	Mouse antibody anti-pan-MUC1 epitope	Murine IgG	Pan-Muc1
CIM301-4	Anti-HIV-gp120 control antibody	Human IgG1	gp120
CIM301-1	Humanized 5E5 anti-MUC1	Human IgG1	Muc1-(S)Tn
CIM301-8	Defucosylated humanized 5E5 anti-MUC1	Human IgG1	Muc1-(S)Tn

## Data Availability

The data presented in this study are available from the corresponding author on reasonable request.

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
