# Peer review of "Defucosylation of Tumor-Specific Humanized Anti-MUC1 Monoclonal Antibody Enhances NK Cell-Mediated Anti-Tumor Cell Cytotoxicity"

_cancers, 2021, doi:10.3390/cancers13112579_

Round 1

Reviewer 1 Report

The study evaluated a humanized anti-Muc1 tumor specific monoclonal antibody and the main aspect of the study focused on enhancement of ADCC by defucosylation of the anti-Muci (Tn/sTn) humanized IgG1 in comparison with its IgG1 counterpart. The concept of better ADCC in afucosylated IgG1 is not new and well documented for many cancer antibody therapeutics such as transtuzumab and rituximab.
 Broad comments: The study relied on mainly flow cytometry assays to evaluate and quantitate several properties of the defucosylated humanized IgG1. Sensitivity and specificity of flow assays often depend on the antibodies used for the detection. The data from flow assay for binding of the humanized anti-Muc1 antibodies are semi-quantitative using MFI and antibody dependent cytotoxicity assay showed marginal activities among comparison groups in comparison with spontaneous killing.
 Specific comments:
1. Labels on X-axis (binding shift) for the flow histograms in Figure 1B-E are not in high quality and visible. More direct demonstration of the antibody specificity to MUC1 (Tn/sTn) is needed such as direct measurement using ELISA or surface plasmon resonance (SPR) method, a standard method for evaluate antibody binding affinities.
2. For degranulation assay in Figure 2, cytokine release should be measured in addition to CD107a expression on NK cells.
3. It is interesting to show CD16 reduction with increased CIM301-8 antibody concentrations in the NK cell cultures (Figure 3). It would be important to determine CD16 protein levels by an alternative method such as WB. Authors also should measure CD16 mRNA transcripts to understand the mechanism of the reduction by the MUC1 antibody treatment.
4. In Figure 4, more cancer cell lines such as T47D and MCF7 breast cancer cell lines (shown Figure 1) in addition to Jurkat cells should be included in the assay panel.
5. In Figure 5, more cancer cell lines should be included in the assay panel.

Author Response

The study evaluated a humanized anti-Muc1 tumor specific monoclonal antibody and the main aspect of the study focused on enhancement of ADCC by defucosylation of the anti-Muci (Tn/sTn) humanized IgG1 in comparison with its IgG1 counterpart. The concept of better ADCC in afucosylated IgG1 is not new and well documented for many cancer antibody therapeutics such as transtuzumab and rituximab.

 Broad comments: The study relied on mainly flow cytometry assays to evaluate and quantitate several properties of the defucosylated humanized IgG1. Sensitivity and specificity of flow assays often depend on the antibodies used for the detection. The data from flow assay for binding of the humanized anti-Muc1 antibodies are semi-quantitative using MFI and antibody dependent cytotoxicity assay showed marginal activities among comparison groups in comparison with spontaneous killing.

Response: We thank the reviewer for thoroughly reviewing the manuscript and for the valuable suggestions for improvement.

Specific comments:

  1. Labels on X-axis (binding shift) for the flow histograms in Figure 1B-E are not in high quality and visible. More direct demonstration of the antibody specificity to MUC1 (Tn/sTn) is needed such as direct measurement using ELISA or surface plasmon resonance (SPR) method, a standard method for evaluate antibody binding affinities.

     Response: We have updated the figures with a high-resolution version. We agree with the reviewer that investigations of the antibody specificity could be of interest for further studies. However, the aim of the present study was to establish the functionality of these humanized anti-MUC1 antibodies against several human cancer cells lines. In follow-up studies, such as xenograft models or using primary human tumor cells, we plan to include these analyses that will also provide insight in possible off-target binding.

  1. For degranulation assay in Figure 2, cytokine release should be measured in addition to CD107a expression on NK cells.

Response: We agree with the reviewer that analysis of cytokines, such as IFN-γ, could provide further insights in the anti-tumor responses mounted by NK cells, as these cytokines will influence the other immune and non-immune cells present in the tumor microenvironment. However, in these in vitro assays, our primary readouts were degranulation and cytotoxicity of NK cells. We show that humanized anti-MUC1 antibodies are strong inducers of NK cell degranulation (Figure 2E-G) and promote cytotoxicity via ADCC (Figure 5). Even though additional experimental analyses cannot be performed at this stage due to time constraints, we plan to include these analyses into cytokine release in future studies.

  1. It is interesting to show CD16 reduction with increased CIM301-8 antibody concentrations in the NK cell cultures (Figure 3). It would be important to determine CD16 protein levels by an alternative method such as WB. Authors also should measure CD16 mRNA transcripts to understand the mechanism of the reduction by the MUC1 antibody treatment.

Response: We appreciate the reviewer’s suggestion for further investigation of the role of our humanized anti-MUC1 antibodies in CD16 shedding. Much work on this subject has been performed by Daniel Davis and colleagues (Srpan et al., 2018; reference 51 in the manuscript; 10.1083/jcb.201712085). In addition, Romee et al. (2013, reference 24 in the manuscript; 10.1182/blood-2012-04-425397) and Peruzzi et al. (2013; reference 25; 10.4049/jimmunol.1300313) provide valuable insights into the dynamics of CD16 shedding in NK cells and the role of CD16 in the engagement of NK cell in serial tumor cell killing. We agree with the reviewer that western blots (or intracellular FACS analysis) and RT-PCRs would be important steps in unraveling the full mechanism of CD16 downregulation, but this is beyond the current scope of the present manuscript.

  1. In Figure 4, more cancer cell lines such as T47D and MCF7 breast cancer cell lines (shown Figure 1) in addition to Jurkat cells should be included in the assay panel.

Response: In our manuscript, we focused on two different cancer cell lines that show high MUC1 expression levels. As shown in Figure 1E, T-47D is the highest MUC1-Tn expressor among the tested three breast cancer cell lines. MCF7 cells express intermediate MUC1-Tn antigen levels which are less attractive as a model to demonstrate functionality of the antibodies. Jurkat cells are included in the assay panel due to high MUC1-Tn epitope expression.

  1. In Figure 5, more cancer cell lines should be included in the assay panel.

Response: In our manuscript, we focused on two different cancer cell lines that show high MUC1 expression levels. As shown in Figure 1E, T-47D cell line is the highest MUC1-Tn expression among the tested three breast cancer cell lines. Cell lines are a useful model for demonstrating functionality of a newly developed antibody, but follow-up studies (e.g. xenograft models and experiments with primary human tumor cells) are required to address the question if this antibody can be used for clinical applications. For example, the previous anti-MUC1 antibody (huHMFG1, AS1402) exert potent anti-tumor response in vitro (Reference 51 Moreno, M., et al. 2007. DOI: 10.1016/j.canlet.2007.06.016), whereas in the phase I (Pegram, M. D., et al. 2009. DOI: 10.1186/bcr2409) and phase II anti metastasis breast cancer clinical trials (Reference 26 Ibrahim, N. K., et al. 2011. DOI: 10.1158/1078-0432.ccr-11-1151) have no impressive outcome. Thus, it is difficult to extrapolate results obtained using cell lines to the clinical setting. For this, patient studies will be required. Therefore, instead of conducting additional in vitro assays with cell lines, we would like to prioritize the planning of patient studies.

Reviewer 2 Report

While Muc1 remains an interesting tumor target, several anti-Muc1 antibodies were tested in clinical studies without success. Gong et al conducted humanization of 5E5, an anti-MUC1 mouse mAb that recognizes MUC1-Tn and MUC1-STn epitopes. The authors confirmed that the humanized antibodies specifically bind to MUC1, increase degranulation of human NK cells, and induce ADCC. The NK cell-mediated anti-tumor responses were enhanced by the defucosylated version of the antibody. Further, the authors reported that certain endocytosis inhibitors improved the bioavailability of MUC1 antigens but did not alter NK cell degranulation or ADCC. They suggested that the humanized antibodies have the potential for therapeutic considerations. While the study was important, there are several concerns need to be addressed.

  1. More details of the antibody humanization should be included.
  2. The Kds of the humanized antibodies binding to MUC1 should be measured and compared to the original 5E5.
  3. The authors may test the efficacies of the humanized antibodies in a mouse model or using primary tumor cells.
  4. Fig 1D: why did 214D4 (anti-pan-MUC1) show much less staining of Jurkat cells than 5E5 and derivatives whereas its staining of other cells was greater than 5E5 based antibodies?

Author Response

While Muc1 remains an interesting tumor target, several anti-Muc1 antibodies were tested in clinical studies without success. Gong et al conducted humanization of 5E5, an anti-MUC1 mouse mAb that recognizes MUC1-Tn and MUC1-STn epitopes. The authors confirmed that the humanized antibodies specifically bind to MUC1, increase degranulation of human NK cells, and induce ADCC. The NK cell-mediated anti-tumor responses were enhanced by the defucosylated version of the antibody. Further, the authors reported that certain endocytosis inhibitors improved the bioavailability of MUC1 antigens but did not alter NK cell degranulation or ADCC. They suggested that the humanized antibodies have the potential for therapeutic considerations. While the study was important, there are several concerns need to be addressed.

Response: We thank the reviewer for thoroughly reviewing the manuscript and for the feedback.

  1. More details of the antibody humanization should be included.

Response: We have changed section 2.1 (Antibodies), providing more details on the humanization of the antibody. As indicated, some of the details of the antibody are covered by a research license with Cancer Research Technology Ltd. and can therefore not be disclosed.

  1. The Kds of the humanized antibodies binding to MUC1 should be measured and compared to the original 5E5.

Response: We currently do not have data on the binding affinity of the humanized anti-MUC1 antibody. However, the Kd for the murine 5E5 antibody has been reported by Sørensen et al. (2005; 10.1093/glycob/cwj044) and Posey et al. (2016; 10.1016/j.immuni.2016.05.014). In our study, we included both antibodies in binding assays using flow cytometry and found comparable levels of binding of the murine 5E5 antibody and the humanized CIM301-1 and CIM301-8 antibodies. Since the aim of the present study was to establish the functionality of these humanized anti-MUC1 antibodies against several human cancer cells lines, we did not investigate this further. In follow-up studies, such as xenograft models or using primary human tumor cells, we plan to include these analyses that will also provide insight in possible off-target binding.

  1. The authors may test the efficacies of the humanized antibodies in a mouse model or using primary tumor cells.

Response: We agree with the reviewer that an important next step in determining the efficacy of these newly generated antibodies could include xenograft studies or in vitro assays using primary human tumor cells. However, the aim of the present study was to establish the functionality of these humanized anti-MUC1 antibodies against several human cancer cells lines, and therefore these assays are planned only for follow-up studies.

  1. Fig 1D: why did 214D4 (anti-pan-MUC1) show much less staining of Jurkat cells than 5E5 and derivatives whereas its staining of other cells was greater than 5E5 based antibodies?

Response: We thank the reviewer for this interesting question. In independent assays, we repeatedly find lower binding levels of the pan anti-MUC1 214D4 antibody compared to anti-MUC1-Tn/STn antibodies (i.e. 5E5, CIM301-1 and CIM301-8). Jurkat cells have been reported to carry a mutation in the COSMC gene (Posey et al., 2016; 10.1016/j.immuni.2016.05.014 and Ju and Cummings, 2002; 10.1073/pnas.262438199) which encodes a chaperone for core 1 beta3-galactosyltransferase (C1beta3Gal-T). We hypothesize that the inability to synthesize normal O-glycans leads to lower expression levels of normally glycosylated MUC1 epitope, which is recognized by the 214D4 antibody. Instead, Jurkat cells consequently exclusively express Tn epitopes which are bound with a higher intensity by 5E5 and CIM301-1 and CIM301-8 than by the pan-MUC1 214 antibody. 

Round 2

Reviewer 1 Report

Author's responses to my comments emphasized the importance of those points that I raised, but the authors argued that those experiments are for the future study and out the scope for this manuscript. I understand author's reluctance for not adding more experiments but it would strengthen the results presented in the manuscript by adding those data set as pointed in my review comments.

Author Response

Comments from Reviewer 1:

Author's responses to my comments emphasized the importance of those points that I raised, but the authors argued that those experiments are for the future study and out the scope for this manuscript. I understand author's reluctance for not adding more experiments but it would strengthen the results presented in the manuscript by adding those data set as pointed in my review comments.

Response: We thank the reviewer for again reviewing our manuscript. We acknowledge that the addition of more experiments as suggested by the reviewer may strengthen the interpretation of our results. However, we also feel that our interpretation of the results is straight-forward. It does not seem that reviewer one disagrees with our conclusion to the interpretation of the dataset, nor in the first review process, neither in this second term. We now have included the suggested assays as proposed by the reviewer in our discussion as a limitation of this study (Lines 582-584, Lines 599-600 and Lines 642).

Reviewer 2 Report

The authors have partially addressed the reviewer's comments. The flow cytometry experiments as shown may not enable to quantitatively compare the antigen bindings of the murine 5E5 antibody and the humanized antibodies. If the antigen binding abilities of the humanized antibodies are much lower than that of the original 5E5, it is unclear whether these new antibodies have sufficient potential for therapeutic applications.

Author Response

Comments from Reviewer 2:

“The authors have partially addressed the reviewer's comments. The flow cytometry experiments as shown may not enable to quantitatively compare the antigen bindings of the murine 5E5 antibody and the humanized antibodies. If the antigen binding abilities of the humanized antibodies are much lower than that of the original 5E5, it is unclear whether these new antibodies have sufficient potential for therapeutic applications”.

Response: We thank the reviewer for reviewing this second version of our manuscript and for providing additional feedback.

 The main finding in the manuscript is the improvement of tumor cell killing mediated by different antibodies via CD16 binding to NK cells. We do support this observation by demonstrating that better antibody binding correlates with better target cell killing. The exact relationship between binding and activation of NK cells is not exactly established, as far as we know. We would like to argue that the hypothesis put forward by the reviewer that our antibody might have lower antigen binding ability is – given the enhanced NK cell-mediated killing – unlikely.

 We have compared the humanized 5E5 antibody amino acid sequence with the murine 5E5. A recent paper reveals the 3D structure of the murine 5E5 antibody binding to the GalNAc-Tn epitope (10.1039/d0cc06349e, now incorporated as reference 51). We found that the proposed crucial amino acids (T100 and F102 in VH as well as Y98 and Y100 in VL) in the 5E5 scFv (in Ref 51), are identical with those in our humanized 5E5 antibody (see below). Based on the observations that our humanized 5E5 antibody binds the MUC1-Tn epitope on multiple cell lines at the same level as the murine 5E5, we hypothesize that the antigen binding affinity of CIM301-1 and CIM301-8 antibodies are similar to the murine 5E5. To confirm this hypothesis, the corresponding assays proposed by reviewer will be performed in future experiments. We have added this argument in the annotate version manuscript (Lines 573-581). Here, we share the alignment results in the CDR3 region of both heavy and light chain with humanized 5E5 CIM301-1/8 and the murine 5E5.

 The reviewer wonders whether these new antibodies have sufficient potential for therapeutic applications since the FACS data may not be optimal for binding studies. Indeed, knowing surface plasma resonance data will add quantitative data to our knowledge and we added this in the discussion at Lines 582-583.

 Thus, we agree with the reviewer on the value of the additional experiments proposed and will perform these in further studies. However, we also argue that the main finding in the manuscript is still valid without this additional experiment.

Round 3

Reviewer 2 Report

The authors stated that they made edits in lines 573-581 and 582-583, which addressed the reviewer's comments. 

Author Response

Comments from Reviewer 2: The authors stated that they made edits in lines 573-581 and 582-583, which addressed the reviewer's comments. 

Response: We thank the reviewer for carefully checking our manuscript again. We previously uploaded the updated manuscript PDF file with tracking-only, however when we prepared the response letter and counted the line number in the version with all changes-tracking-version. That is the reason the line number is different. We have uploaded the correct PDF version now. We feel sorry to have made this mistake. Please check the current version, with correct indicated line numbers.
